# Enhancing birth weight outcomes through improved antenatal care in West African countries: Evidence from propensity score matching analysis

Aklilu Habte Hailegebireal[1,2]*, Habtamu Mellie Bizuayehu[3], Samuel Hailegebreal Gele[4,5], Angwach Abrham Asnake[6]

**1** School of Public Health, College of Medicine and Health Sciences, Wachemo University, Hosanna, Ethiopia, **2** Faculty of Health and Environmental Sciences, School of Clinical Sciences, Auckland University of Technology, Auckland, New Zealand, **3** First Nations Cancer and Wellbeing (FNCW) Research Program, School of Public Health, The University of Queensland, Brisbane, Queensland, Australia, **4** Department of Health Informatics, College of Medicine and Health Sciences, School of Public Health, Wachemo University, Hosanna, Ethiopia, **5** Menzies School of Health Research, Charles Darwin University, Darwin, Northern Territory, Australia, **6** Department of Epidemiology and Biostatistics, School of Public Health, College of Health Sciences and Medicine, Wolaita Sodo University, Wolaita Sodo, Ethiopia

* akliluhabte57@gmail.com

## Abstract

While many underlying causes of low birth weight (LBW, < 2500 grams) are preventable through overall antenatal care (ANC) uptake, limited empirical evidence exists in West Africa quantifying the impact of the WHO-recommended four or more visits (ANC4+) and eight or more visits (ANC8+) models using rigorous causal inference techniques. Thus, this study aimed to examine the causal effect of those two models on birth weight outcomes in West African countries using propensity score matching (PSM). This study included a weighted sample of 51,455 women sourced from Demographic and Health Surveys (DHS) data from 14 West African countries (2012–2023). Covariates associated with treatments (ANC4+ and ANC8+) and birth weight were identified using Chi-square tests, one-way analysis of variance (ANOVA), and t-tests, and those deemed confounders were included in PSM. PSM was applied using a logit model to estimate the causal effects of ANC4+ and ANC8+ on birthweight. Average Treatment Effect on the Treated (ATT) and Average Treatment Effect (ATE) were calculated for each exposure. Quality of matching was assessed both statistically and graphically. Sensitivity analysis was done using Rosenbaum bounds to estimate unmeasured confounding and confirm the robustness of treatment effects. The mean birthweight was 3121.2 grams (±669.0), with 10.42% of newborns classified as LBW. Receiving ANC8+ and ANC4 + were associated with a birthweight increase of 102.36 and 83.89 grams, respectively. Across the entire population, ANC8+ and ANC4 + were linked to average birthweight increases of 89.09 and 75.81 grams, respectively. Of the weighted sample, 71.38% and 14.54% of women

**Data availability statement:** The data used in this study were obtained from the Demographic and Health Surveys (DHS) Program. These datasets are publicly available in anonymised form upon reasonable request via the DHS website: https://dhsprogram.com/data/avail-able-datasets.cfm. Anyone can access the data by registering and submitting a brief request outlining the intended use, in accordance with DHS data access policies. The specific datasets used in this study include (list of 14 country-years, IR files).

**Funding:** The authors received no specific funding for this work.

**Competing interests:** The authors have declared that no competing interests exist.

received ANC4+ and ANC8+, respectively. All matching diagnostics demonstrated strong covariate balance and confirmed the validity of the treatment effect estimates. The treatment and control groups were well comparable for the baseline confounders after matching (p-value > 0.05). This study found that receiving ANC4+ or ANC8+ has a positive effect on birth weight, underscoring the importance of scaling up efforts to ensure comprehensive ANC coverage, especially ANC8+, in the region to reduce LBW prevalence and improve neonatal survival.

## Introduction

The birth weight of a newborn is the initial weight documented immediately after birth, preferably assessed within the first hours post-delivery [1]. The World Health Organization (WHO) defines low birth weight (LBW) as a birth weight of less than 2,500 grams [1,2].

Recent estimates showed that 19.8 million liveborn newborns (14.7% of all live births) globally were LBW [3]. The majority (91%) of these LBW births took place in low- and middle-income countries (LMICs), with almost half (48%) and a quarter (24%) of them occurring in southern Asia and sub-Saharan Africa, respectively [4].

LBW is associated with increased morbidity, mortality, and disability during child-hood, and those who survived faced long-term consequences such as physical and neurological developmental delays, disabilities, and chronic health conditions, including type 2 diabetes, hypertension, and cardiovascular disease [5–13]. LBW babies were more likely to die during their first month of life [5,8–13].

Several maternal, fetal, and environmental factors contribute to LBW. Of fetal factors, preterm birth and intrauterine growth restriction are common causes of LBW. Maternal and environmental factors include maternal undernutrition [14,15], infections during pregnancy (e.g., malaria, polyparasitism, COVID-19) [16–18], maternal anaemia [19,20], hypertensive disorders [21–23], placental insufficiency [24,25], and multiple gestation [26,27]. Behavioral factors such as maternal smoking [28–30] and alcohol or substance use [31–33] further exacerbate the risk of LBW.

Many of those underlying causes and risk factors for LBW are largely preventable or manageable through the receipt of timely and adequate antenatal care (ANC) [34–37]. ANC paves the way for receiving key services for a healthy pregnancy, which can help to prevent, identify, and treat the conditions that cause LBW and thus foster achievement of the World Health Assembly (WHA) nutrition target to reduce LBW by 30% between 2012 and 2030 [38,39]. It also plays a pivotal role in address-ing underlying determinants of LBW, such as maternal malnutrition, undiagnosed health conditions, and limited health education [40]. The WHO originally endorsed the four-visit ANC model (ANC4+) as the recommended approach for antenatal care. In 2016, however, this was revised, with the new guidelines stipulating at least eight ANC (ANC8+) contacts as the minimum requirement [41,42].

Overall, equity in ANC coverage is vital for global public health because unequal access disproportionately affects vulnerable populations, contributing to higher rates

of LBW, a key predictor of neonatal mortality and long-term health risks [38]. Thus, ensuring equitable ANC access and quality remains a strategic pathway for countries, especially in high-burden regions like West Africa, to achieve meaningful improvements in birth outcomes and meet international nutrition targets [38,39].

LBW continues to be a critical public health issue in West Africa, where uneven access to maternal healthcare leads to a heightened burden of LBW [43,44], contributing to persistent inequities in neonatal survival and long-term health outcomes. However, there is a dearth of empirical evidence linking ANC adequacy to birthweight. Especially, the evidence delineating the relative effects of ANC4+ versus ANC8+ remains limited, leaving a gap in understanding how the shift toward WHO's new standard translates into measurable health benefits, such as reducing LBW. While some studies have reported associations between ANC quality and birth weight [40,45–47], little is known with robust methods like propensity score matching (PSM) to estimate the actual causal effect.

This study addresses that evidence gap by exploring the impact of ANC adequacy on birth weight in West Africa, where access to quality ANC is often uneven [48]. The findings directly support global efforts to reduce preventable neonatal morbidity and mortality, align with Sustainable Development Goals (SDGs), and advance equity by informing policies that prioritize underserved communities in low-resource contexts. The analysis uses PSM to address confounding and selection bias inherent in observational data, allowing a robust comparison of the effect of the two ANC models (ANC4+ and ANC8+). Insights from this research carry important implications for tailoring ANC policies, enhancing service delivery strategies, and guiding resource allocation toward the most effective coverage benchmarks. Ultimately, findings could contribute to progress towards the Global Nutrition Target, which aims to reduce LBW by 30% by 2025, a goal now extended to 2030 due to slow progress.

## Methods

### Study design, period, and population

This study utilizes appended Demographic and Health Surveys (DHS) data from 14 West African countries, collected between 2012 and 2023. The analysis draws on the Individual Record (IR) files, which include women who had given birth within the five years preceding each survey. Women without documented data on ANC and/or birth weight were excluded from the study. We considered countries that had conducted DHS reports from 2010 onward, and we got countries with surveys conducted between 2012 and 2023. The final selection yielded a weighted sample of 51,455 eligible women, providing a robust foundation for the analysis (Table 1).

**Sampling techniques and procedures.** The DHS employs a two-stage stratified sampling design: first selecting enumeration areas (EAs) with probability proportional to size (PPS), followed by systematic random sampling of households within selected EAs. Eligible women from each home were interviewed using a structured and standardized questionnaire. The details are mentioned in the Demographic and Health Survey Sampling and Household Listing Manual issued by ICF International [49,50].

### Measurement of variables of the study

Outcome variable:

The outcome variable is birthweight, which is the initial weight recorded immediately after birth, preferably within the first hours post-delivery, prior to any substantial postnatal weight reduction [1]. It is a continuous variable and recorded in grams as variable m19-1, indicating either recorded measurements from health cards or maternal recall. The analysis includes only children born within the five years preceding the survey who had valid birthweight data, excluding entries with standard placeholders used in the dataset to indicate specific types of missing or implausible responses (coded as 9996, 9997, or 9998). Using birthweight in continuous form allows for a more precise estimate of the effect of treatment (ANC4+ and ANC8+), enhances statistical power, and avoids information loss from categorizing the variable [51,52].

**Table 1. List of countries included in the analysis with their respective weighted sample size, 2013–2023.**

| Country | Year | Study participants(Weigted Sample) | Percent (%) |
|---|---|---|---|
| Burkina Faso | 2021 | 5,320 | 10.34 |
| Benin | 2018 | 5,802 | 11.28 |
| Ivory Coast | 2021 | 4,129 | 8.02 |
| Ghana | 2021 | 3,754 | 7.29 |
| Gambia | 2020 | 4,344 | 8.44 |
| Guinea | 2018 | 2,832 | 5.5 |
| Liberia | 2020 | 1,354 | 2.63 |
| Mali | 2018 | 2,567 | 4.99 |
| Mauritania | 2019 | 1,438 | 2.79 |
| Nigeria | 2018 | 5,704 | 11.08 |
| Niger | 2012 | 2,176 | 4.23 |
| Sierra Leone | 2019 | 5,017 | 9.75 |
| Senegal | 2023 | 3,948 | 7.67 |
| Togo | 2014 | 3,070 | 5.97 |
| Total1 | 14 | 51,455 | 100.00 |

**The exposure (treatment) variable.** Antenatal care (ANC) recorded as m14-1 is the main treatment variable, which is operationalised based on the number of visits a woman received during her last pregnancy. This discrete numeric variable captures the total ANC contacts, with values above 20 considered implausible and excluded. For analysis purposes, we grouped the treatment variable into two key groups: four or more visits (ANC4+) and eight or more visits (ANC8+), aligning with the previous [53,54] and current [41,42] WHO recommendations, respectively. Women with eight or more ANC visits were coded as Yes = 1; those with fewer were coded as No = 0. Similarly, women with four or more ANC visits were coded as Yes = 1; those with fewer were coded as No = 0. Two distinct models were fitted: Model 1 examined the causal effect of ANC8+ on birth weight, while Model 2 assessed the causal effect of ANC4+ on birth weight. The control groups for model 1 and model 2 were women without ANC8+ and ANC4 +, respectively.

**Explanatory variables.** Given the observational nature of the DHS, random assignment of exposure was not applied, leading to inherent differences between treatment and control groups. To address potential selection bias in estimating the effects of ANC4+ and ANC8+ on birthweight, we identified maternal baseline characteristics that influence both the exposures and the outcome (birth weight) based on prior literature [27,41] (Table 2).

**Confounding variables.** Those variables significantly associated with both the exposures (ANC8+ and ANC4+) and birthweight were treated as confounders and were incorporated into the Propensity Score Matching (PSM). Those variables are maternal age, level of education, head of household, husband occupation, women's occupation, place of residence, wealth index, family size, parity, contraceptive uptake, perceived distance to a health facility, ease of getting money for health care, permission to go to the health facility, listening to the radio, watching Television, reading newspaper, timing of first ANC visit, and visiting a health facility within the last 12 months. The hypothesized causal relationships among confounders, exposures (ANC4+ and ANC8+), and birthweight outcomes were illustrated using a Directed Acyclic Graph (DAG), developed in DAGitty version 3 (Fig 1).

## Statistical analysis

Data management and statistical analyses were conducted using STATA version 18. All analyses were weighted using the DHS-provided sampling weight ($\frac{v005}{1000000}$) to ensure representativeness of the sample. We run chi-square tests to assess the association between the exposure variables (ANC8+ and ANC4+) and categorical covariates. In addition, one-way

**Table 2. Potential predictors of adequate ANC and birth weight.**

| Variables | Description and categorization |
| --- | --- |
| Age | The number of completed years from the date of birth to the date of interview which further grouped as 15–29, 20–34, and 35–49 years. |
| Marital status | Self-reported legal or consensual union status of the woman aged 15–49 years at the time of interview, which was categorized as married, never married, and formerly married (divorced, widowed, and separated). |
| Educational status | The highest level of schooling attended or completed by the respondents, which is categorised as no education, primary, secondary, and higher education. |
| Residence | The type of place where the respondent lives, based on administrative classification, was categorized as urban or rural. |
| Family size | Total count of people who lived in the household, which was categorised as ≥5 and <5 |
| Head of Household | The person identified by household members as the head, often responsible for decision-making, is categorised by sex as male or female. |
| Parity | The number of living children the woman had at the time of the survey was categorized into nulliparous (0), primiparous (1), multiparous (2–4), and grand multiparous (≥5). |
| Preceding birth interval | The number of months between the birth of a child and the birth of the immediately preceding sibling, which is further categorized as First birth (which has no birth interval), <12 months, 12–23 months, and ≥24months |
| Contraceptive uptake | A percentage of women whether they or their partner are currently using any method to delay or avoid getting pregnant and categorized as (i) *non-user*, *(ii)hormonal contraceptives* [injectables, implants, pills], and *non-hormonal* [Intrauterine contraceptive device, female condom, male condom, Diaphragm, permanent methods (tubal ligation, vasectomy), lactational amenorrhea method, standard days method, any traditional method. |
| Timing of first antenatal check | The number of months pregnant a woman was when she received her first ANC visit during her most recent pregnancy, further categorised as early (first ANC visit in 1st trimester (≤3 months)) and late first ANC visit in 2nd or 3rd trimester (>3 months). |
| Media exposure | Number of women aged 15–49 who are exposed to specific media with various frequencies: read a newspaper, watch television, and listen to the radio, and categorized as: Not at all, less than once a week, and at least once a week |
| Wealth index | A composite measure used to assess a household's economic status based on asset ownership and housing characteristics. It is calculated using principal components analysis (PCA) to assign weights to items such as flooring material, access to electricity, ownership of durable goods, and sanitation facilities. Each household is given a standardized score, which is then used to rank all households and divide them into five quintiles: poorest, poorer, middle, richer, and richest. |
| Iron intake | Self-reported consumption of iron-containing supplements during the most recent pregnancy, as Yes or No |
| Pregnancy intention | Self-report of women about their intention toward their most recent pregnancy, which is further categorised as wanted then (planned), wanted later (mistimed), or not wanted at all (unwanted) |
| Cigarette smoking | Indicates current smoking status, which is further classified as a smoker if she reports smoking cigarettes at the time of the interview (=Yes), and as a non-smoker if she does not (No=0). |
| Problems in accessing healthcare | Measured based on women's self-reported barriers, including obtaining permission to seek care, getting money for treatment, distance to a health facility, and not wanting to go alone, the response to each of those items was categorised as "big problem" or "not a big problem". |

**PLOS** **Global Public Health**

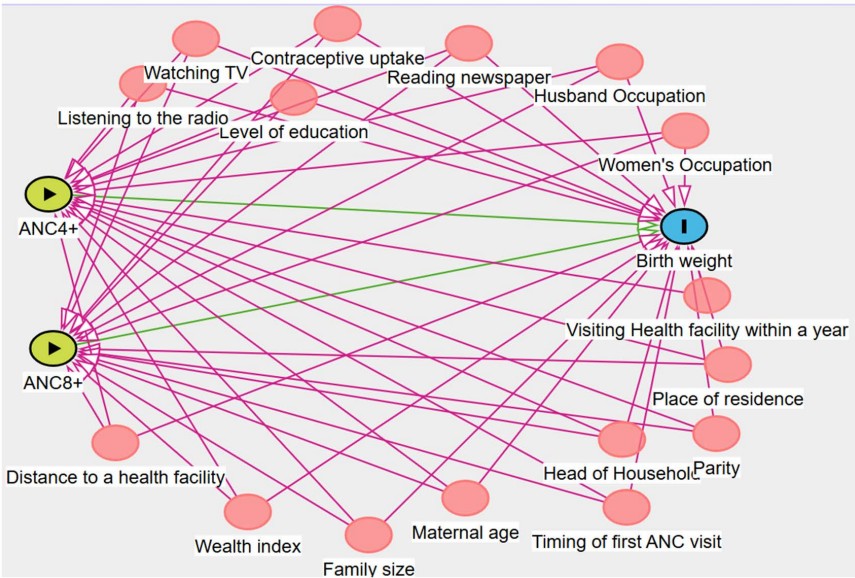

**Fig 1. A directed acyclic graph to show the relationship between treatment, outcome, and confounders.**

analysis of variance (ANOVA) and independent t-tests were used to assess the association between covariates and birth weight, to identify possible confounders for Propensity Score Matching (PSM). Accordingly, the observed covariates were classified into three categories based on their relationship with the exposure and outcome variables: those associated solely with the exposure assignment, those linked to both the exposure and the outcome—considered confounders—and those associated exclusively with the outcome (birth weight). Only covariates identified as confounders were incorporated into the propensity score model in this study.

**Propensity score matching.** This study used data from DHS, an observational dataset in which treatment and control groups were not assigned randomly, resulting in baseline disparities and potential mismatches in confounding variables [55,56]. Such imbalances could bias the estimation of the causal effects of adequate antenatal care (ANC 8+ and ANC4+) on birth weight. Thus, a PSM was done to create a balanced comparison between treated and untreated groups, reducing confounding and improving the validity of the causal effect [57,58]. The matching was undertaken for each treatment using a logit model, and propensity scores (the conditional probability of receiving treatments-ANC4+ and ANC8+) were estimated using the pscore command. The propensity score is the likelihood of receiving treatments given all confounders, and that ranges from 0 to 1 [56,57].

After creating the matched samples, the effects of ANC4+ and ANC8+were assessed independently by comparing the difference in birth weight (in grams) between the treated and control groups. Using logistic regression via the *psmatch2* command, the study matched newborn birth weight to maternal exposure to ANC4+ and ANC8+, enabling estimation of treatment effects within balanced comparison groups. The use of the logit model is due to the exposures being dichotomous (Yes = 1 or No = 0). In addition, covariate balance before and after matching was assessed using the pstest command, with p-values below 0.05 interpreted as evidence of imbalance. Conversely, a covariate with a p-value greater than 0.05 after matching is interpreted as evidence of adequate balance between the treatment and control groups [59–61].

We seek to estimate the average effect of receiving treatments on birth weight independently. Let $Y_i^T$ denote the birth weight for the $i^{th}$ newborn whose mother received the treatments (either ANC4+ or ANC8+), and $Y_i C$ represents the hypothetical birth weight for the same newborn had the mother not received the respective treatments. The treatment

assignment indicator T equals 1 if the mother received ANC8+, and 0 otherwise. The observed outcome can thus be described as: $Y_i = T_i Y_{i1} + (1 - T_i) Y_{i0}$ [62,63] Where

$Y_i$: observed birth weight for individual i, $T_i$: treatment indicator (1 if exposed to treatment (ANC4+ or ANC8+, 0 otherwise), $Y_{i1}$: potential outcome if treated (exposed), and $Y_{i0}$: potential outcome if not treated (unexposed)

The individual treatment effect is $Y$ $Y_{i1}-Y_{i0}$, but since we can never observe both outcomes for the same individual, our interest shifts to the average effect of treatment on the treated (ATT):

ATT = $E[Y_{i1}- Y_{i0}/T_i = 1]$ [62,63], where

$Y_{i1}$: birth weight if the mother received ANC8+ (the treatment), $Y_{i0}$: birth weight if the same mother had not received treatments (ANC8+ or ANC4+) (the counterfactual), and $T_i$: treatment indicator (1 if the mother received treatments, 0 if not)

Covariates were selected before treatments to minimize post-treatment bias. Key PSM assumptions, including common support and unconfoundedness (selection on observables), were evaluated both graphically and statistically. Matching was restricted to treated mothers whose propensity scores for birth weight fell within the range of controls. Nearest neighbor (with and without replacement) and radius matching (caliper = 0.01) were assessed using *pstest* for balance diagnostics (to ensure a high degree of similarity between matched units while retaining sufficient sample size). Treatment effects (ATT, ATU, ATE) were estimated via psmatch2, with the final model selected based on matching quality.

The quality of matching was evaluated using multiple diagnostics to ensure robustness of the propensity score model. These included assessments of standardized bias to examine covariate balance, model significance tests, and evaluation of the common support assumption. Additionally, a post-matching covariate balance plot was employed to visually inspect improvements in balance across groups (The details are mentioned in the results section).

To assess the robustness of the estimated treatment effect to potential unobserved confounding, a sensitivity analysis was conducted using Rosenbaum bounds (*rosbounds*) [64,65] for continuous outcomes, quantifying how strongly an unmeasured variable would need to influence treatment assignment to nullify the observed effect [57,66,67]. The sensitivity parameter Gamma (Γ) was used to represent the degree of departure from random assignment; values of Γ greater than 1 indicate increasing levels of potential hidden bias. The threshold at which the treatment effect becomes statistically insignificant provides insight into the strength of evidence for causality under unmeasured confounding [57,65].

**Ethical approval and consent to participate.** This study is based on secondary analysis of publicly available data from the MEASURE DHS program. At the time of access, the dataset was available and unrestricted under the data use policies in effect prior to recent changes introduced by the current U.S. government. Ethical approval and informed consent were not required, as the dataset is fully anonymized and contains no identifiable personal or household information. Formal permission to download and use the data was granted through the DHS Program website, http://www.dhsprogram.com.

## Results

### Baseline characteristics of the study population

This analysis is based on a weighted sample of 51,455 women across 14 West African countries, with the largest proportions coming from Benin (11.28%), Nigeria (11.08%), and Burkina Faso (10.34%). The mean age of respondents was 29.26 years (±7.07), and nearly half were in the age group 25–34 years. The majority (44.7%) of women had no formal education, yet the vast majority were in a marital relationship (89.88%). Notably, over a quarter belonged to the richest wealth quintile. More than half (53.33%) of women were multiparous (Table 3).

The uptake of adequate ANC and mean birthweight showed significant variation across socioeconomic characteristics. Younger women (15–24) were less likely to reach ≥8 visits compared to those aged 35–49. ANC utilization increased with educational attainment; only 7.2% of women with no education had ≥8 visits, compared to 40.8% among those with higher education (p < 0.001). Urban residence, higher wealth, smaller family size, and female-headed households were all

**Table 3.** The distribution of sociodemographic characteristics of respondents across adequacy of ANC (ANC4+ and ANC8+) and birth weight in West African countries, 2012-2023.

| Variable categories | Total (Weighted frequency, N=51,455) | ≥4 ANC visit | | | ≥8 visits | | | Birth weight (weighted estimates) | |
|---|---|---|---|---|---|---|---|---|---|
| | | No(%) | Yes(%) | p-value* | No(%) | Yes(%) | p-value* | Mean birthweight (g) (95%CI) | p-value** |
| **Country** | | | | | | | | | |
| Burkina Faso | 5,320(10.34) | 25.66 | 74.34 | <0.001 | 99.09 | 0.91 | <0.001 | 2998.7(2982.7, 3014.7) | <0.001*** |
| Benin | 5,802(11.28) | 40.12 | 59.88 | | 89.39 | 10.61 | | 3048.8(3032.9, 3064.7) | |
| Ivory Coast | 4,129(8.02) | 35.78 | 64.22 | | 95.25 | 4.75 | | 3082.3(3051.7, 3112.85) | |
| Ghana | 3,754(7.29) | 7.64 | 92.36 | | 57.48 | 42.52 | | 3082.0(3056, 3108.0) | |
| Gambia | 4,344(8.44) | 20.78 | 79.22 | | 95.56 | 4.44 | | 3112.7(3091.6, 3133.9) | |
| Guinea | 2,832(5.5) | 52.48 | 47.52 | | 95.48 | 4.52 | | 3332.7(3299.1, 3366.3) | |
| Liberia | 1,354(2.63) | 8.32 | 91.68 | | 70.11 | 29.89 | | 3221.7(3144.0, 3299.4) | |
| Mali | 2,567(4.99) | 39.13 | 60.87 | | 94.27 | 5.73 | | 3321.9(3281.6, 3362.2) | |
| Mauritania | 1,438(2.79) | 48.47 | 51.53 | | 94.23 | 5.77 | | 2861.1(2808.7, 2913.5) | |
| Nigeria | 5,704(11.08) | 14.06 | 85.94 | | 55.48 | 44.52 | | 3310.5(3288.2, 3332.9) | |
| Niger | 2,177(4.23) | 54.49 | 45.51 | | 99.78 | 0.22 | | 3043.6(3019.8, 3067.4) | |
| Sierra Leone | 5,017(9.75) | 19.51 | 80.49 | | 78.32 | 21.68 | | 3180.1(3163.5, 3196.7) | |
| Senegal | 3,948(7.67) | 28.55 | 71.45 | | 92.20 | 7.80 | | 3054.2(3023.4, 3085) | |
| Togo | 3,070(5.97) | 31.74 | 68.26 | | 95.84 | 4.16 | | 3153.8(3128.8, 3178.8) | |
| **Age** | | | | | | | | | |
| 15-24 | 14,023(27.25) | 31.9 | 68.12 | <0.001 | 89.4 | 10.6 | <0.001 | 3037.2 (3026.4, 3048.0) | 0.020*** |
| 25-34 | 24,569(47.75) | 27.0 | 73.0 | | 84.12 | 15.9 | | 3143.2 (3135.0, 3151.6) | |
| 35-49 | 12,864(25) | 28.2 | 71.81 | | 83.7 | 16.2 | | 3173.0 (3161.5, 3184.6) | |
| **Educational level** | | | | | | | | | |
| No education | 22,963(44.63) | 36.2 | 63.83 | <0.001 | 92.84 | 7.23 | <0.001 | 3095.0 (3085.3, 3104.9) | <0.001*** |
| Primary | 9,443(18.35) | 30.1 | 69.91 | | 89.0 | 11.0 | | 3116.1 (3098.8, 3133.4) | |
| Secondary | 15,626(30.37) | 20.3 | 79.72 | | 78.30 | 21.71 | | 3168.9 (3154.7, 3183.1) | |
| Higher | 3,423(6.65) | 11.5 | 88.21 | | 59.21 | 40.83 | | 3239.4 (3210.5, 3268.3) | |
| **Marital status** | | | | | | | | | |
| Unmarried | 3,430(6.67) | 26.9 | 73.11 | 0.126 | 83.32 | 16.74 | | 3083.0(3048.7, 3117.3) | 0.002*** |
| Married | 46,250(89.88) | 28.7 | 71.32 | | 85.73 | 14.33 | | 3135.0(3127.6, 3142.4) | |
| Others[a] | 1,775(3.45) | 29.5 | 70.50 | | 84.21 | 15.82 | | 3118.6(3075.4, 3161.8) | |
| **Residence** | | | | | | | | | |
| Urban | 25,475(49.51) | 25.3 | 74.71 | <0.001* | 80.91 | 19.12 | <0.001 | 3156.1(3144.7, 3167.4) | <0.001 |
| Rural | 25,981(50.49) | 31.9 | 68.10 | | 89.90 | 10.12 | | 3106.3(3097.5, 3115.2) | |
| **Wealth index** | | | | | | | | | |
| Poorest | 7,440(14.46) | 34.40 | 65.60 | <0.001* | 91.62 | 8.38 | <0.001 | 3070.3(3054.54, 3086.2) | <0.001*** |
| Poorer | 8,845(17.19) | 31.86 | 68.14 | | 90.70 | 9.30 | | 3087.5(3071.7, 3103.3) | |
| Middle | 9,926(19.29) | 31.30 | 68.70 | | 88.36 | 11.64 | | 3112.3(3097.01, 3127.6) | |
| Richer | 11,560(22.47) | 28.16 | 71.84 | | 84.64 | 15.36 | | 3143.9(3127.9, 3159.8) | |
| Richest | 13,685(26.6) | 21.84 | 78.16 | | 77.33 | 22.67 | | 3194.6(3178.9, 3210.31) | |
| **Family size** | | | | | | | | | |
| ≤5 members | 20,106(39.08) | 25.44 | 74.56 | <0.001* | 79.79 | 20.21 | <0.001 | 3144.5(3132.6, 3156.3) | 0.014** |
| >5 members | 31,349(60.92) | 30.66 | 69.34 | | 89.10 | 10.90 | | 3122.3(3113.30, 3131.3) | |

*(Continued)*

**Table 3.** (Continued)

| Variable categories | Total (Weighted frequency, N=51,455) | ≥4 ANC visit | | | ≥8 visits | | | Birth weight (weighted estimates) | |
|---|---|---|---|---|---|---|---|---|---|
| | | No(%) | Yes(%) | p-value* | No(%) | Yes(%) | p-value* | Mean birthweight (g) (95%CI) | p-value** |
| **Head of household** | | | | | | | | | |
| Male | 41,712(81.07) | 29.24 | 70.76 | <0.001* | 86.09 | 13.91 | <0.001 | 3135.1(3127.6, 3143.4) | 0.004*** |
| Female | 9,743(18.93) | 25.99 | 74.01 | | 82.78 | 17.22 | | 3111.4(3094.1, 3128.85) | |

a widowed, divorced, separated.

p-value* for chi-square test.

P-value** for an independent t-test.

p-value *** for one-way ANOVA.

associated with better ANC coverage. Birthweight increased progressively with maternal age, education, urban residence, and parity (p<0.001). The highest average birth weights were observed in Guinea and Mali, while Mauritania had the lowest (Table 3).

### Adequate ANC uptake and birth weight across health service-related characteristics

Almost half (49.32%) of women initiated ANC early (within the first trimester). More than three-quarters (78.67%) of pregnancies were wanted at the time, though 17.2% were mistimed and 4.13% were unwanted. Over half of the women (51.76%) reported financial barriers as a major challenge to accessing maternity care, while about one-third faced issues related to distance (31.19%) (Table 4). The uptake of adequate ANC and neonatal birthweight showed significant variation across obstetric characteristics. Receipt of ANC4+varied by parity, with primiparous women exhibiting the highest uptake at 73.64%, closely followed by multiparous women (72.49%). Notably, women who initiated their first ANC contact during the first trimester demonstrated markedly higher uptake, with ANC4+(84.27%) and ANC8+(20.69%). Multiparous and grand multiparous women exhibited greater mean birthweights of 3157.8 and 3163.1 grams, respectively, compared with nulliparous women (2901.7 grams) (Table 4).

### Adequate ANC uptake and birthweight in the region

The weighted prevalence of women who received at least four antenatal care visits (ANC4+) was 71.38% (95% CI: 70.9–71.7), whereas only 14.54% (95% CI: 14.15–14.92) received eight or more visits (ANC8+). The mean(±SD) birthweight was 3121.2 grams (±669.0), with 10.42% (95% CI: 10.10, 10.74) of newborns classified as LBW (<2500 g). Coverage of ANC4+was highest in Ghana (92.36%), Liberia (91.68%), and Nigeria (85.94%). Similarly, Nigeria (44.52%), Ghana (42.52%), and Liberia (29.89%) reported the greatest proportions of women receiving ANC8+ (Fig 2).

**Estimation of propensity scores of treatment.** The variables that have a significant association with both treatments (ANC8+ and ANC4+) and outcome (birthweight) were considered for matching using the logit model (Table 5).

**Propensity score of receiving treatments.** For the treatment ANC4+, the overall propensity score was 0.71, indicating a moderate level of variability in the likelihood of receiving treatment across the sample. Which means women in the sample had a 71% predicted probability of receiving treatment, given their covariates. This is supported by the histogram (Fig 3). Whereas the average estimated propensity score for receiving ANC8+was 0.14, indicating that, based on observed covariates, the overall likelihood of receiving ANC8+ in the study population was 14%. The low average propensity score likely reflects the low proportion of women in the general population receiving ANC8+ (14.53%). The histogram below illustrates the distribution of propensity scores by treatment status, highlighting baseline differences in treatment probability between groups (Fig 4).

**Table 4. The distribution of health service-related characteristics of respondents across ANC4+, ANC8+, and birth weight in West African countries.**

| Variable categories | Total (Weighted frequency, N = 51,455) | ≥4 ANC visit (%) | | | ≥8 visit(%) | | | Birth weight(95%CI) (weighted estimates) | |
|---|---|---|---|---|---|---|---|---|---|
| | | No | Yes | p-value | No | Yes | p-value | Mean birthweight (gm) | p-value |
| **Parity** | | | | | | | | | |
| Nulliparous | 296(0.57) | 34.82 | 65.18 | <0.001* | 90.24 | 9.76 | <0.001* | 2901.7(2779.9, 3023.5) | <0.001*** |
| Primiparous | 12,577(24.44) | 26.36 | 73.64 | | 83.81 | 16.19 | | 3049.2(3034.2, 3064.1) | |
| Multiparous | 27,444(53.33) | 27.51 | 72.49 | | 84.21 | 15.79 | | 3157.8(3148.0, 3167.7) | |
| Grand multiparous | 11,138(21.65) | 33.77 | 66.23 | | 90.29 | 9.71 | | 3163.1(3148.4, 3177.9) | |
| **Timing of ANC** | | | | | | | | | |
| Early | 25,380(49.32) | 15.73 | 84.27 | <0.001* | 79.31 | 20.69 | <0.001* | 3131.7(3121.1, 3142.5) | 0.317** |
| Late | 26,075(50.68) | 41.18 | 58.82 | | 91.45 | 8.55 | | 3130.2(3120.6, 3139.8) | |
| **Pregnancy status** | | | | | | | | | |
| Wanted then | 40,479(78.67) | 27.92 | 72.08 | <0.121 | 85.50 | 14.50 | 0.072 | 3131.1(3123.1, 3139.0) | 0.091*** |
| Wanted later | 8,852(17.2) | 31.47 | 68.53 | | 85.99 | 14.01 | | 3118.5(3099.9, 3137.0) | |
| Wanted no more | 2,124(4.13) | 30.19 | 69.81 | | 83.59 | 16.41 | | 3180.7(3143.8, 3217.6) | |
| **Preceding birth interval** | | | | | | | | | |
| First birth | 12,076(23.47) | 26.24 | 73.76 | 0.132 | 83.99 | 16.01 | 0.081 | 3034.6(3019.3, 3049.9) | 0.213*** |
| <12 months | 238(0.46) | 34.44 | 65.56 | | 85.42 | 14.58 | | 3159.5(3002.7, 3236.3) | |
| 12-23 months | 4,782(9.29) | 35.33 | 64.67 | | 84.73 | 15.27 | | 3162.5(3138.4, 3186.6) | |
| >=24 months | 34,359(66.78) | 28.49 | 71.51 | | 86.09 | 13.91 | | 3160.5(3151.9, 3169.1) | |
| **Contraceptive uptake** | | | | | | | | | |
| Non-users | 36,550(71.03) | 30.85 | 69.15 | <0.001* | 86.90 | 13.10 | <0.001* | 3123.6(3115.2, 3132.1) | 0.001** |
| Users | 14,905(28.97) | 23.16 | 76.84 | | 81.95 | 18.05 | | 3149(3135.4, 3162.5) | |
| **Pregnancy termination** | | | | | | | | | |
| Yes | 7,824(15.21) | 24.64 | 75.36 | 0.112 | 82.39 | 17.61 | 0.161 | 3154.1(3134.4, 3173.8) | 0.112** |
| No | 43,631(84.79) | 26.34 | 73.66 | | 84.02 | 15.98 | | 3146.1(3119.1, 3134.5) | |
| **Smoking cigarette** | | | | | | | | | |
| Yes | 559(1.09) | 30.92 | 69.08 | 0.156 | 89.17 | 10.83 | 0.111 | 3020.9(2956.4, 3085.4) | 0.147** |
| No | 50,896(98.91) | 28.60 | 71.40 | | 85.42 | 14.58 | | 3132.1(3124.9, 3139.4) | |
| Iron intake | | | | | | | | | |
| Yes | 48,215(93.70) | 27.99 | 72.01 | 0.048 | 85.68 | 14.32 | 0.259 | 3131.7(3125.7, 3137.7) | 0.408** |
| No | 3,240(6.30) | 47.07 | 52.93 | | 82.31 | 17.69 | | 3120.4(3095.6, 3145.2) | |
| **Perceived distance** | | | | | | | | | |
| Big problem | 16,046(31.19) | 30.76 | 69.24 | <0.001* | 87.91 | 12.09 | <0.001* | 3115.2(3102.5, 3127.8) | 0.003** |
| Not a big problem | 35,409(68.81) | 27.66 | 72.34 | | 84.36 | 15.64 | | 3138.1(3129.3, 3146.9) | |
| **Getting permission** | | | | | | | | | |
| Big problem | 9,373(18.22) | 33.33 | 66.67 | <0.001* | 89.40 | 10.60 | <0.001* | 3137.8(3119.9, 3155.9) | 0.2113** |
| Not a big problem | 42,082(81.78) | 27.58 | 72.42 | | 84.59 | 15.41 | | 3129.4(3121.6, 3137.3) | |
| **Getting money** | | | | | | | | | |
| Big problem | 26,633(51.76) | 30.82 | 69.18 | <0.001* | 87.87 | 12.13 | <0.001* | 3114.8(3105.1, 3124.5) | <0.001** |
| Not a big problem | 24,822(48.24) | 26.27 | 73.73 | | 82.88 | 17.12 | | 3148.7(3137.7, 3159.0) | |
| **Autonomy in decision making** | | | | | | | | | |
| Non-autonomous | 16,150(31.39) | 31.92 | 68.08 | <0.001* | 90.82 | 9.18 | 0.131 | 3133.4(3125.0, 3142.8) | 0.145** |
| Autonomous | 35,305(68.61) | 28.15 | 71.85 | | 84.04 | 15.96 | | 3138.8(3125.0, 3142.8) | |

*(Continued)*

**Table 4.** (Continued)

| Variable categories | Total (Weighted frequency, N = 51,455) | ≥4 ANC visit (%) | | | ≥8 visit(%) | | | Birth weight(95%CI) (weighted estimates) | |
|---|---|---|---|---|---|---|---|---|---|
| | | No | Yes | p-value | No | Yes | p-value | Mean birthweight (gm) | p-value |
| **Watching TV** | | | | | | | | | |
| Not all | 21,320(41.43) | 33.65 | 66.35 | | 89.63 | 10.37 | <0.001* | 3098.9(3081.2, 3098.4) | <0.001*** |
| <once a week | 9,095(17.68) | 29.62 | 70.38 | | 85.32 | 14.68 | | 3133.9(3119.7, 3148.2) | |
| At least once | 21,040(40.89) | 23.10 | 76.90 | | 81.31 | 18.69 | | 3171.3(3161.8, 3180.8) | |
| **Listening to the radio** | | | | | | | | | |
| Notat all | 19,343(37.59) | 31.55 | 68.45 | <0.001* | 87.51 | 12.49 | <0.001* | 3101.7(3092.3, 3111.1) | <0.001*** |
| <once a week | 13,391(26.02) | 28.12 | 71.88 | | 85.39 | 14.61 | | 3130.3(3119.1, 3141.6) | |
| At least once a week | 18,721(36.38) | 25.96 | 74.04 | | 83.41 | 16.59 | | 3161.7(3151.7, 3171.6) | |
| **Reading newspaper** | | | | | | | | | |
| Notat all | 45,521(88.47) | 29.72 | 70.28 | <0.001* | 86.77 | 13.23 | <0.001* | 3119.2(3113.0, 3125.3) | <0.001*** |
| <once a week | 3,875(7.53) | 20.24 | 79.76 | | 74.99 | 25.01 | | 3209.9(3186.4, 3233.5) | |
| At least once a week | 2,059(4.0) | 20.13 | 79.87 | | 76.34 | 23.66 | | 3243.5(3211.8, 3273.1) | |

p-value* for chi-square test.

P-value** for an independent t-test.

p-value *** for one-way ANOVA.

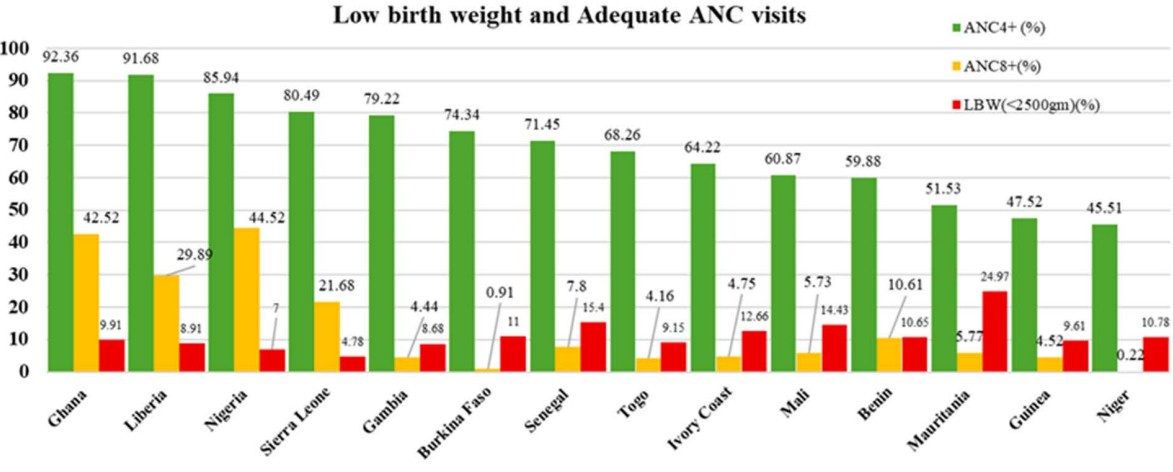

**Fig 2. Low birth weight and the uptake of adequate ANC (ANC4+ and ANC8+) in West African countries.**

**The causal effect of adequate ANC (ANC4+ and ANC8+) on birth weight.** The causal effects of two ANC interventions, ANC4+ and ANC8 +, on birth weight were assessed by estimating the mean difference in birth weight between the treated groups, those who received ANC4+ or ANC8 +, and their matched counterparts who received inadequate care (fewer than four or eight visits, respectively). The PSM analysis estimates the effect of the ANC4+ and ANC8+ on birth weight by adjusting for potential confounders (Table 6).

PSM was performed to estimate the Average Treatment Effect on the Treated (ATT) by comparing women who received the treatment (ANC4+ and ANC8+) with a matched group of similar women without treatment, based on observed covariates. It also estimates the Average Treatment Effect (ATE), reflecting the treatments' impact across the

**Table 5. Results of logit regression analysis of factors associated with treatments (ANC4+ and ANC8+) in West African countries.**

| Variables | Treatment 1 (ANC4+) | | Treatment 2 (ANC8+) | |
|---|---|---|---|---|
| | Coefficient | p-value | Coefficient | p-value |
| Maternal age | 0.2019249 | <0.001 | 0.3291493 | <0.001 |
| Level of education | 0.3743412 | <0.001 | 0.5877336 | <0.001 |
| Head of HH | 0.054766 | 0.05 | 0.1084411 | 0.001 |
| Husband Occupation | 0.0819643 | 0.042 | 0.4937964 | <0.001 |
| Women's occupation | 0.3450422 | <0.001 | 0.7869784 | <0.001 |
| Residence | 0.0498653 | 0.057 | -0.1289432 | <0.001 |
| Wealth index | -0.0307419 | 0.002 | 0.0223269 | 0.101 |
| Family size | -0.1307413 | <0.001 | -0.4570839 | <0.001 |
| Parity | -0.1149288 | <0.001 | -0.0921337 | <0.001 |
| Contraceptive uptake | 0.1988207 | <0.001 | 0.086108 | 0.003 |
| Perceived distance to a health facility | -0.0714415 | 0.006 | -0.1629254 | <0.001 |
| Ease of getting money for health care | 0.0494267 | 0.041 | 0.0696046 | 0.029 |
| Permission to go to the health facility | 0.2080569 | <0.001 | 0.2299781 | <0.001 |
| Listening to the radio | 0.0428634 | <0.001 | -0.021195 | 0.211 |
| Watching TV | 0.0858998 | <0.001 | -0.0456165 | 0.014 |
| Reading newspaper | -0.1234608 | <0.001 | -0.0990745 | <0.001 |
| Timing of first ANC visit | -1.360186 | <0.001 | -1.05657 | <0.001 |
| Visiting a health facility within the last 12 months | -0.2909038 | <0.001 | | |
| Constant | 2.415923 | <0.001 | -1.971597 | <0.001 |

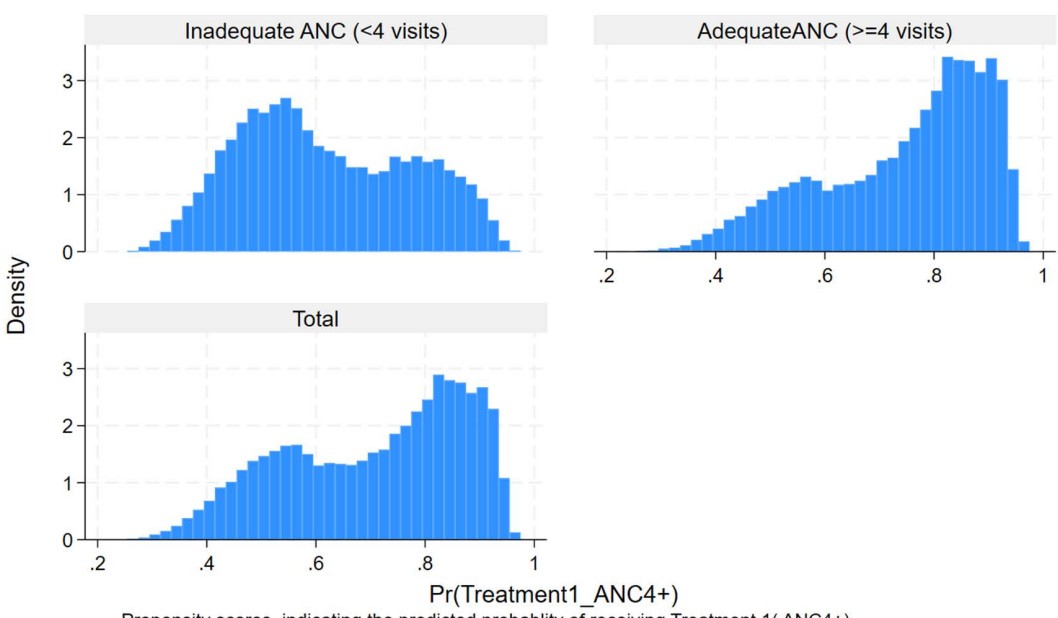

**Fig 3. Histogram showing the distribution of estimated propensity scores by treatment status (ANC4+vs. fewer visits).**

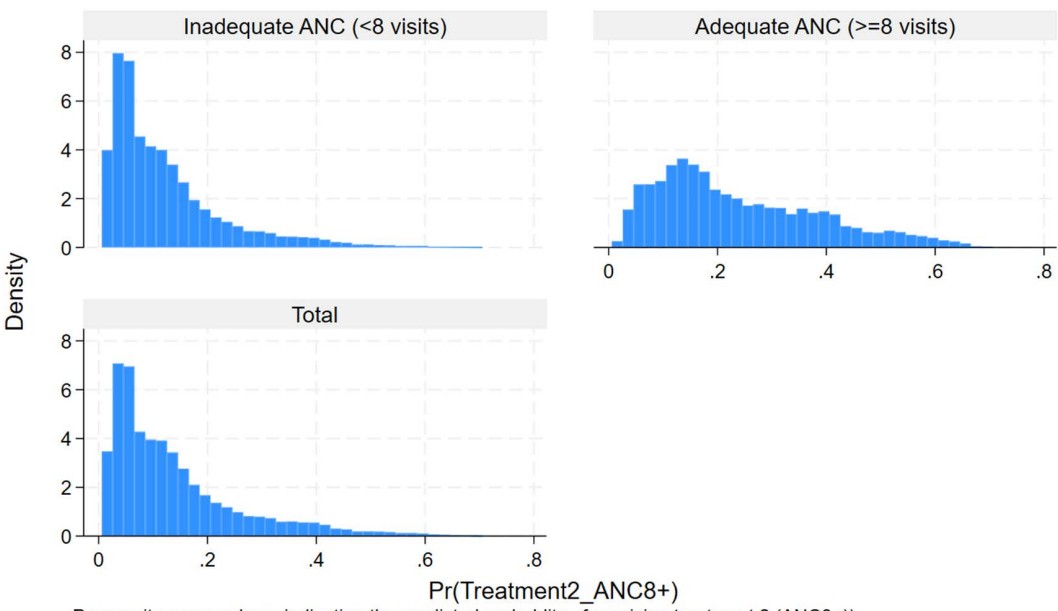

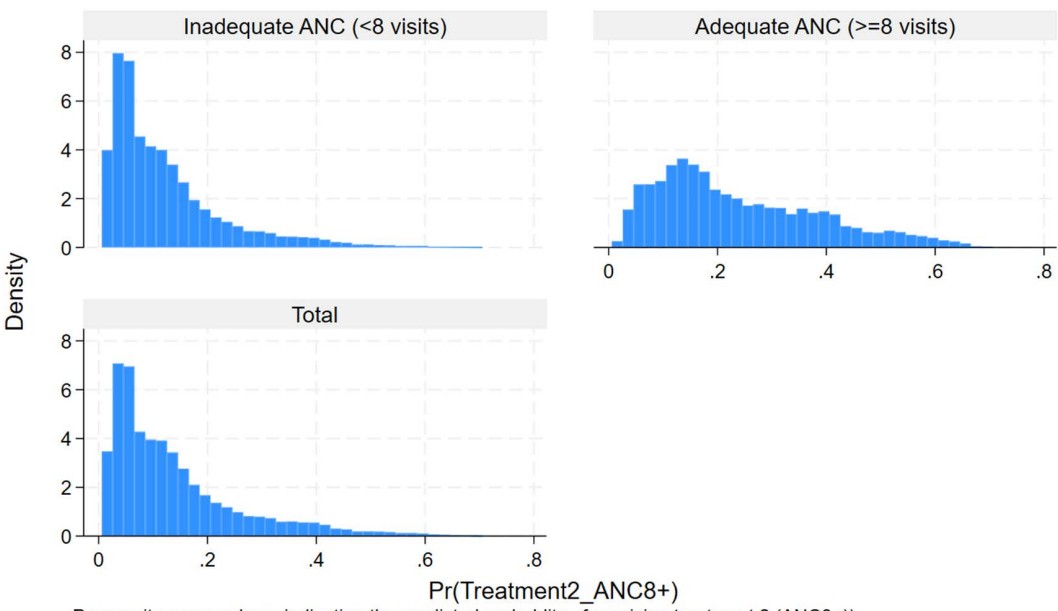

**Fig 4. Histogram showing the distribution of estimated propensity scores by treatment status (ANC8 + vs. fewer visits).**

**Table 6. A propensity score-matched analysis of the impact of Treatments (ANC4+ and ANC8+) on birth weight in West African countries, 20102-2023.**

| Treatments | Sample | Treated | Controls | Difference (mean) | S.E. | T-stat |
|---|---|---|---|---|---|---|
| ANC8+ | Unmatched | 3234.07 | 3103.15 | 130.92 | 8.46 | 15.46 |
| | ATT | 3234.067 | 3131.70 | 102.36 | 13.30 | 7.69 |
| | ATU | 3103.15 | 3168.70 | 67.68 | . | . |
| | ATE | | | 82.32 | . | . |
| ANC4+ | Unmatched | 3147.40 | 3057.71 | 89.69 | 6.42 | 13.96 |
| | ATT | 3147.40 | 3063.50 | 83.89 | 11.33 | 7.40 |
| | ATU | 3057.70 | 3113.9 | 56.20 | . | . |
| | ATE | | | 75.81 | . | . |

Key: ATT: Average Treatment Effect among Treated, ATU: Average Treatment Effect on Untreated, ATE: Average Treatment Effect on the whole population.

entire women, including both treated and untreated groups. A radius matching approach with a 0.01 caliper width was employed to optimize match quality and estimate the average treatment effect (ATE) across the overall population, both exposed and unexposed groups.

Accordingly, the unmatched estimates indicated that receiving eight or more ANC visits (ANC8+) was associated with increases in birth weight of 130.92 grams. The ATE estimates of 89.09 showed that having ANC8 + was associated with increases in birth weight of 89.09 grams across the entire population. The ATT for ANC8 + was 102.36, which implies that after adjusting for confounders using PSM, women having ANC8 + had newborns with significantly higher birth weights by 102.36 grams compared to their matched counterparts (women who received fewer than eight visits). The Average Treatment Effect on the Untreated (ATU) was calculated to be 67.68 grams, implying that if women who did not receive ANC8 + had gotten the intervention, their newborns' birth weights would have increased by about 67.68 grams.

Likely, the unmatched estimates indicated that receiving four or more antenatal care visits (ANC4+) was linked with increases in birth weight of 89.69 grams. The ATT was 83.89, which implies that after adjusting for confounders using PSM, women who received ANC4 + had newborns with significantly higher birth weights by 83.89 grams compared to their matched counterparts who received fewer than four visits. The ATE estimates showed that receiving ANC4 + was 75.81grams across the entire population. The Average Treatment Effect on the Untreated (ATU) was 56.20, meaning that if women lacking ANC4 + had received the intervention, the birth weight would have increased by an estimated 56.2 grams (Table 6).

**Quality of matching.  Standard bias and model significance:** The ability of the matching procedure to balance the distribution of relevant covariates between the treatment and control groups was evaluated. The pseudo-$R^2$ statistic was used to assess how well the covariates explained the likelihood of treatment assignment. For the first treatment (ANC8+), the pseudo-$R^2$ declined markedly from 0.134 in the unmatched sample to 0.001 post-matching, indicating a significant reduction in covariate imbalance between treatment and control groups. Additionally, the mean standardized bias decreased from 25.6% to 1.7%, while the median bias reduced from 19.1% to 01.4%, collectively demonstrating substantial improvement in covariate balance and suggesting that the matching procedure was successful. For the other treatment (ANC4+), the pseudo-$R^2$ value decreased markedly from 0.11 before matching to 0.001 post-matching, while the maximum absolute standardized bias (B) dropped from 84.8% to 7.3%, indicating a significant improvement in covariate balance (Table 7).

* Ratio of Treated vs Untreated group.

**Balancing test:** The pstest command in Stata was employed to assess covariate balance between treated and control groups before and after matching. Before matching, most covariates exhibited substantial bias. Differences between the unmatched and matched samples were assessed using t-tests. The t-tests indicated there was a significant mean difference for each covariate before matching ($p < 0.05$), but after matching, the percentage bias for all covariates was reduced, and the results indicated no significant mean differences across nearly all factors ($p > 0.05$). In both cases, the matching process decreased the bias by more than 95% for each covariate. This demonstrates that the variables were adequately balanced (Tables 8 and 9).

**Common support assumption:** The common support assumption was assessed graphically and which revealed substantial overlap in the distribution of propensity scores between treated and control groups, substantiating the validity of the common support assumption. Visual inspection of the graph for ANC4 + revealed the propensity score of both treated and control groups concentrated between 0.4 and 0.8. This pattern suggests that a majority of individuals had a good predicted probability of receiving ANC4 +, and also, substantial overlap was evident, supporting the presence of common support within the matched sample (Fig 5). Visual inspection of the graph for ANC8 + revealed a markedly skewed distribution of estimated propensity scores, with both treated and control groups concentrated at the lower end of the scale (0–0.3) and tapering off toward higher values. This pattern suggests that a majority of individuals had a low predicted probability of receiving ANC8 +, yet substantial overlap was still evident, supporting the presence of common support within the matched sample (Fig 6).

**Table 7. Performance of the propensity score matching for the effect of ANC8+ and ANC4+ on birthweight.**

| Intervention | Sample | Ps R2 | LR chi2 | p > chi2 | MeanBias | MedBias | B | R* | %Var |
|---|---|---|---|---|---|---|---|---|---|
| ANC8+ | Unmatched | 0.134 | 5602.81 | <0.001 | 25.6 | 19.1 | 100.4* | 0.90 | 81 |
| | Matched | 0.001 | 28.46 | 0.049 | 1.7 | 1.4 | 8.9 | 1.02 | 6 |
| ANC4+ | Unmatched | 0.110 | 6943.81 | <0.001 | 17.1 | 13.7 | 84.8* | 1.22 | 97 |
| | Matched | 0.001 | 99.61 | 0.32 | 1.1 | 1.1 | 7.3 | 1.10 | 17 |

**Table 8. Quality of matching for the effect of ANC4+ on birthweight in the West African region, 2012-2023.**

| Variable | | Mean | | %bias | Bias reduction | T-test | |
|---|---|---|---|---|---|---|---|
| | | Treated | | | Control | T_statstics | p-value |
| Maternal age | Unmatched | 1.9857 | 1.9445 | 5.6 | | 5.87 | <0.001 |
| | Matched | 1.9857 | 1.9892 | -0.5 | 91.5 | -0.66 | 0.511 |
| Level of education | Unmatched | 1.0425 | .66675 | 39.8 | | 40.11 | <0.001 |
| | Matched | 1.0425 | 1.0322 | 1.1 | 97.3 | 2.03 | 0.043 |
| Head of HH | Unmatched | 1.19 | 1.1612 | 7.6 | | 7.77 | <0.001 |
| | Matched | 1.19 | 1.187 | 0.8 | 89.8 | 1.02 | 0.306 |
| Husband Occupation | Unmatched | .93244 | .92416 | | 3.2 | 3.37 | 0.001 |
| | Matched | .93244 | 0. 93317 | -0.3 | 91.2 | -0.40 | 0.692 |
| Women's occupation | Unmatched | .72279 | .64558 | 16.7 | | 17.57 | <0.001 |
| | Matched | .72279 | .7283 | -1.2 | 92.9 | -1.68 | 0.093 |
| Residence | Unmatched | 1.5305 | 1.583 | -10.6 | | -10.99 | <0.001 |
| | Matched | 1.5305 | 1.5392 | -1.8 | 83.5 | -2.37 0 | 0.018 |
| Wealth index | Unmatched | 3.1981 | 2.985 | 15.1 | | 15.70 | <0.001 |
| | Matched | 3.1981 | 3.1928 | 0.4 | 97.5 | 0.50 | 0.615 |
| Family size | Unmatched | 1.6116 | 1.6673 | -11.6 | | -12.01 | <0.001 |
| | Matched | 1.6116 | 1.6272 | -3.3 | 92.4 | -4.38 | <0.001 |
| Parity | Unmatched | 2.9553 | 3.0258 | -10.0 | | -10.47 | <0.001 |
| | Matched | 2.9553 | 2.9673 | -1.7 | 93.0 | -2.35 | 0.019 |
| Preceding birth interval | Unmatched | 1.2353 | 1.2665 | -6.0 | | -6.30 | <0.001 |
| | Matched | 1.2353 | 1.2178 | 3.4 | 95.4 | 4.82 | 0.047 |
| Contraceptive uptake | Unmatched | .30513 | .22758 | 17.6 | | 17.96 | <0.001 |
| | Matched | .30513 | .30567 | -0.1 | 99.3 | -0.16 | 0.873 |
| Planning status of last pregnancy | Unmatched | 1.2353 | 1.2665 | -6.0 | | -6.30 | <0.001 |
| | Matched | 1.2353 | 1.2178 | 3.4 | 94.0 | 4.82 | 0.521 |
| Perceived distance to access maternity services | Unmatched | 1.6877 | 1.6474 | 8.6 | | 8.96 | <0.001 |
| | Matched | 1.6877 | 1.6912 | -0.7 | 96.4 | -1.02 | 0.409 |
| Accessing money for health care | Unmatched | 1.493 | 1.4309 | 12.5 | | 12.95 | <0.001 |
| | Matched | 1.493 | 1.4912 | 0.4 | 97.0 | 0.50 | 0.617 |
| Permission to go to the health facility | Unmatched | 1.8334 | 1.8305 | 0.7 | | 1.06 | 0.288 |
| | Matched | 1.493 | 1.4912 | 0.4 | 97.0 | 0.50 | 0.617 |
| Watching TV | Unmatched | .99627 | .80552 | 21.3 | | 21.99 | 0.000 |
| | Matched | .99627 | .9893 | 0.8 | 96.3 | 1.04 | 0.297 |
| Listening to the radio | Unmatched | 1.0039 | .8845 | 13.9 | | 14.39 | <0.001 |
| | Matched | 1.0039 | 1.0026 | 0.2 | 98.8 | 0.22 | 0.828 |

**Post-matching covariate balance plot:** To evaluate the effectiveness of the PSM procedure for women who received eight or more antenatal care (ANC8+) visits, covariate balance was assessed using standardized percentage bias plots. This visual diagnostic compared pre- and post-matching standardized biases across key covariates. The resulting graph demonstrated a marked reduction in bias, with all post-matching values tightly clustered around zero and falling well within the conventional ±10% threshold. These findings confirm that the matching algorithm successfully eliminated systematic differences between treatment and control groups across observed covariates, thereby enhancing the validity of subsequent treatment effect estimates for ANC8+ on birthweight outcomes (Fig 7).

**Table 9. Quality of matching for the effect of ANC8+ on birthweight in the West African region, 2012-2023.**

| Variable | | Mean | | %bias | Bias reduction | T-test | |
|---|---|---|---|---|---|---|---|
| | | Treated | | | Control | T-statstics | p-value |
| Maternal age | Unmatched | 2.07 | 1.9582 | 15.6 | | 12.11 | <0.001 |
| | Matched | 2.07 | 2.0688 | 0.2 | 98.9 | 0.11 | 0.915 |
| Level of education | Unmatched | 1.5024 | .84157 | 66.3 | | 54.17 | <0.001 |
| | Matched | 1.5024 | 1.4929 | 1.0 | 98.6 | 0.55 | 0.582 |
| Head of HH | Unmatched | 1.2281 | 1.1741 | 13.5 | | 11.04 | <0.001 |
| | Matched | 1.2281 | 1.2184 | 2.4 | 92.0 | 1.40 | 0.162 |
| Husband Occupation | Unmatched | .9616 | .92497 | 15.9 | | 11.34 | 0.001 |
| | Matched | .9616 | .96645 | -2.1 | 96.8 | 1.56 | 0.692 |
| Women's occupation | Unmatched | .8396 | .67794 | 38.5 | | 28.03 | <0.001 |
| | Matched | .8396 | .84833 | -2.1 | 94.6 | -1.45 | 0.093 |
| Residence | Unmatched | 1.4308 | 1.5642 | -26.9 | | -21.23 | <0.001 |
| | Matched | 1.4308 | 1.4266 | 0.8 | 96.9 | 0.50 | 0.614 |
| Wealth index | Unmatched | 3.5383 | 3.0714 | 33.5 | | 26.20 | <0.001 |
| | Matched | 3.5383 | 3.5347 | 0.3 | 99.2 | 0.16 | 0.876 |
| Family size | Unmatched | 1.4816 | 1.6513 | -34.7 | | -27.87 | <0.001 |
| | Matched | 1.4816 | 1.4876 | -1.2 | 96.5 | -0.72 | 0.474 |
| Parity | Unmatched | 2.8924 | 2.9893 | -14.2 | | -10.92 | <0.001 |
| | Matched | 2.8924 | 2.8884 | 0.6 | 95.8 | 0.36 | 0.716 |
| Preceding birth interval | Unmatched | 3.1375 | 3.2205 | -6.6 | | -5.25 | <0.001 |
| | Matched | 3.1375 | 3.1396 | -0.2 | 97.5 | -0.10 | 0.923 |
| Perceived distance to access a health facility | Unmatched | 1.7273 | 1.6677 | 13.0 | | 10.04 | <0.001 |
| | Matched | 1.7273 | 1.7324 | -1.1 | 97.4 | -0.69 | 0.488 |
| Accessing money for health care | Unmatched | 1.5568 | 1.4618 | 19.1 | | 15.03 | <0.001 |
| | Matched | 1.5568 | 1.568 | -2.3 | 97.8 | -1.36 | 0.617 |
| Permission to go to the health facility | Unmatched | 1.8707 | 1.8092 | 16.8 | | 12.56 | 0.288 |
| | Matched | 1.8707 | 1.8737 | -0.8 | 95.0 | -0.55 | 0.583 |
| Watching TV | Unmatched | 1.1288 | .91045 | 24.3 | | 19.07 | 0.000 |
| | Matched | 1.1288 | 1.1357 | -0.8 | 96.8 | -0.46 | 0.644 |
| Listening to the radio | Unmatched | 1.0641 | .95387 | 12.8 | | 10.06 | <0.001 |
| | Matched | 1.0641 | 1.055 | 1.0 | 91.8 | 0.63 | 0.728 |

Similarly, a covariate balance was assessed for ANC4 + , and the resulting balance plot demonstrated a marked reduction in standardized mean differences for all included covariates, with post-matching values falling well within the ± 10% threshold(-6–2). This indicates that the matching procedure successfully mitigated baseline differences, supporting the validity of subsequent treatment effect estimates on birthweight (Fig 8).

**Sensitivity analysis:** In the presence of unobserved variables exerting simultaneous influence on both the assignment into exposure and the outcome variable, the emergence of 'hidden bias' becomes a concern. To handle this, the Rosenbaum bounding method was employed (as the outcome is a continuous variable) to ascertain the extent to which unmeasured variables, or hidden bias, impact the selection process and, consequently, the implications of the matching analysis. Strong evidence that ANC8+ and ANC4 + increase neonatal birthweight would be found in all of the analyses, in a study free of bias, that is, where $\Gamma = 1$. The upper bound on the significance level for $\Gamma = 1.05, 1.1, 1.15……2$. The final output demonstrated consistent robustness of the estimated treatment effects across both treatment groups (ANC8+ and

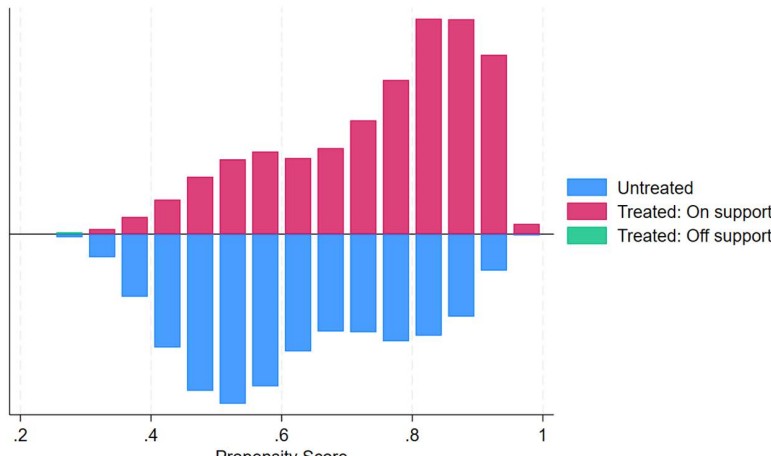

**Fig 5. Histogram of propensity score distribution across treatment (ANC4+) and control groups.**

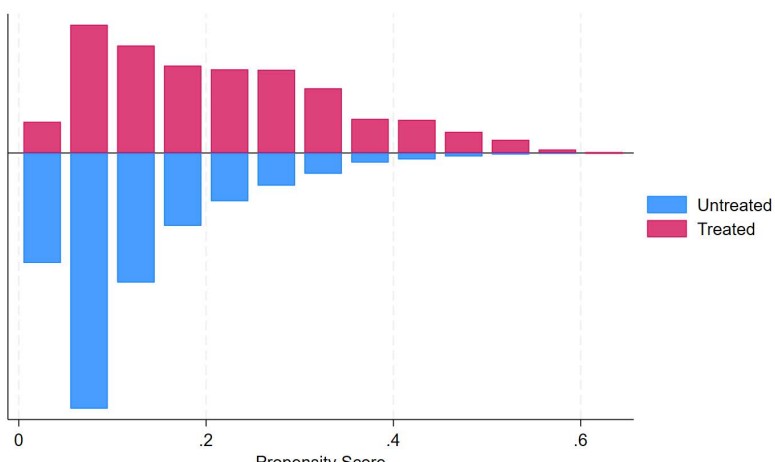

**Fig 6. Histogram of propensity score distribution across treatment (ANC8+) and control groups.**

ANC4+). For each specification, the upper and lower bounds remained statistically significant across a range of Γ values, suggesting that the observed associations are unlikely to be explained by unmeasured confounding, which enhances the credibility of the findings (Tables 10 and 11).

## Discussion

This study investigates the causal relationship between adequate antenatal care—defined by ANC4+ and ANC8 + contacts—and birthweight outcomes in West African countries, using PSM. Propensity score matching is widely recognized as a robust method for causal inference in observational studies, as it facilitates the construction of a comparable control group in the absence of randomization, thereby reducing selection bias [63,68].

The study revealed that 71.38% and 14.54% of women received ANC4+ and ANC8 +, respectively, and the weighted prevalence of LBW was 10.42%. Receiving ANC8 + was associated with an average increase of 102.37 grams (ATT = 102.37), while ANC4 + contacts led to an 83.90 grams (ATT = 83.9) gain compared to their respective control

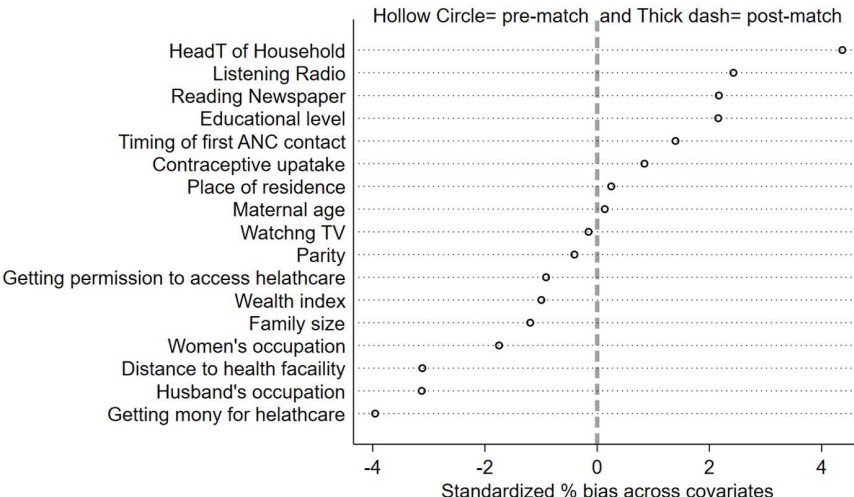

**Fig 7. Standardized percentage bias in the distribution of confounders before and after matching for ANC8+.**

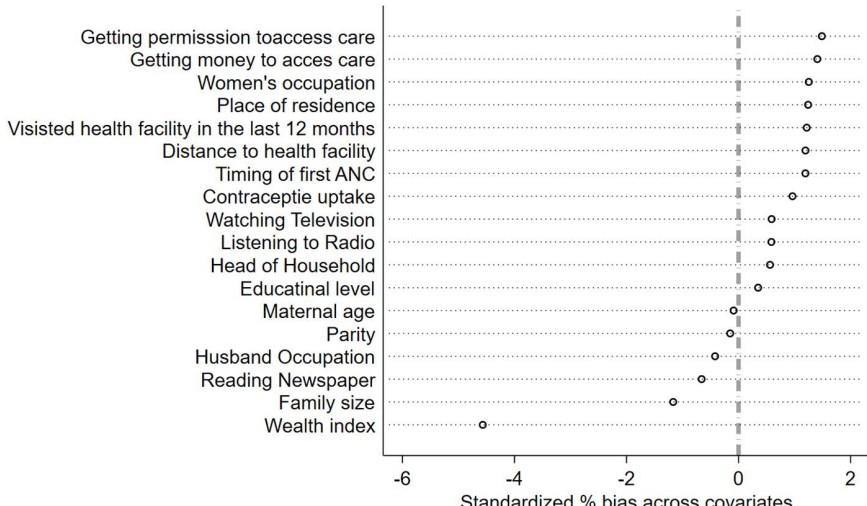

**Fig 8. Standardized percentage bias in the distribution of confounders before and after matching for ANC4+.**

groups. These findings were comparable to the previous studies [35,69,70]. For example, a study conducted between 2019 and 2020 among 8,362 Gambian DHS newborns revealed that each additional ANC visit was linked to a 22-gram increase in birthweight, highlighting the incremental value of sustained maternal care throughout pregnancy [69]. The current finding was also in line with a recent systematic review and meta-analysis in Africa [35]. The birthweight increment associated with ANC4+ or ANC8+ could likely stem from enhanced maternal monitoring, early identification and management of pregnancy-related complications, and increased access to vital interventions, including nutritional counseling and supplementation, which facilitate fetal growth and weight gain during pregnancy [71,72]. In addition, as the number of ANC visits increases, so does the opportunity to detect and manage underlying conditions that contribute to maternal anaemia, such as nutritional deficiencies (by iron, folate), helminthic infections (by deworming), and endemic diseases like malaria, thereby improving birthweight outcomes [73].

**Table 10. Sensitivity analysis using Rosenbaum bounds for birthweight for ANC8+.**

| Gamma | sig+ | sig- | t-hat+ | t-hat- | CI+ | CI- |
|---|---|---|---|---|---|---|
| 1 | 0.00 | 0.00 | 3100 | 3100 | 3100 | 3100 |
| 1.05 | 0.00 | 0.00 | 3065 | 3,100 | 3050 | 3100 |
| 1.1 | 0.00 | 0.00 | 3050 | 3,100 | 3050 | 3100 |
| 1.15 | 0.00 | 0.00 | 3050 | 3,100 | 3050 | 3125 |
| 1.2 | 0.00 | 0.00 | 3050 | 3,150 | 3050 | 3150 |
| 1.25 | 0.00 | 0.00 | 3025 | 3,150 | 3000 | 3150 |
| 1.3 | 0.00 | 0.00 | 3000 | 3,150 | 3000 | 3150 |
| 1.35 | 0.00 | 0.00 | 3000 | 3150 | 3000 | 3150 |
| 1.4 | 0.00 | 0.00 | 3000 | 3175 | 3000 | 3200 |
| 1.45 | 0.00 | 0.00 | 3000 | 3200 | 3000 | 3200 |
| 1.5 | 0.00 | 0.00 | 3000 | 3200 | 3000 | 3200 |
| 1.55 | 0.00 | 0.00 | 3000 | 3200 | 3000 | 3200 |
| 1.6 | 0.00 | 0.00 | 3000 | 3200 | 3000 | 3200 |
| 1.65 | 0.00 | 0.00 | 2990 | 3217.5 | 2970 | 3250 |
| 1.7 | 0.00 | 0.00 | 2950 | 3250 | 2950 | 3250 |
| 1.75 | 0.00 | 0.00 | 2950 | 3250 | 2950 | 3250 |
| 1.8 | 0.00 | 0.00 | 2950 | 3250 | 2950 | 3250 |
| 1.85 | 0.00 | 0.00 | 2950 | 3250 | 2950 | 3250 |
| 1.9 | 0.00 | 0.00 | 2950 | 3250 | 2950 | 3250 |
| 1.95 | 0.00 | 0.00 | 2950 | 3250 | 2925 | 3250 |
| 2 | 0.00 | 0.00 | 2925 | 3250 | 2900 | 3250 |

gamma - log odds of differential assignment due to unobserved factors.

sig+- upper bound significance level.

sig- - lower bound significance level.

t-hat+- upper bound Hodges-Lehmann point estimate.

t-hat- - lower bound Hodges-Lehmann point estimate.

CI+ - upper bound confidence interval (a=.95).

CI- - lower bound confidence interval (a=.95).

Overall, the findings underscore the added value of expanding antenatal care coverage, particularly through the WHO-recommended ANC8+ model, which contributes to improved birthweight outcomes in the region. However, the observed lower rate of both ANC4+ (71.38%) and ANC8+ (14.54%) in the current study highlights a significant gap in optimal coverage. Given the demonstrated measurable benefits of ANC uptake at the population level, enhancing service uptake not only supports better birth outcomes but also reinforces equity-driven approaches by targeting vulnerable groups who are disproportionately affected by LBW. Therefore, integrating ANC expansion into existing health system reforms could serve as a pivotal strategy to reduce LBW and advance neonatal health equity across the region. These findings align with global health priorities, including the Sustainable Development Goal (SDG) 3.2, which aims to end preventable deaths of newborns and children under five by 2030 [74–76]. Low birthweight is a major contributor to neonatal mortality, and its reduction via optimal prenatal care is critical to achieving this target.

The prevalence of LBW in this study was 10.42%, which is notably lower than the recent global estimate of 14.7% [3]. It aligns closely with findings from sub-Saharan Africa (10.44%) [77], though it exceeds another regional estimate of 9.76% [27]. In contrast, the LBW rate observed here is substantially lower than the average reported across Asian countries (16.64%) [76], and significantly lower than country-specific figures such as Pakistan (18%), Afghanistan (14%),

**Table 11. Sensitivity analysis using Rosenbaum bounds for the effect of ANC8+ on birthweight.**

| Gamma | sig+ | sig- | t-hat+ | t-hat- | CI+ | CI- |
|---|---|---|---|---|---|---|
| 1 | 0 | 0 | 3049 | 3049 | 3024 | 3049 |
| 1.05 | 0 | 0 | 2999 | 3049 | 2999 | 3049 |
| 1.1 | 0 | 0 | 2999 | 3049 | 2999 | 3054 |
| 1.15 | 0 | 0 | 2999 | 3074 | 2999 | 3099 |
| 1.2 | 0 | 0 | 2999 | 3099 | 2999 | 3099 |
| 1.25 | 0 | 0 | 2999 | 3099 | 2999 | 3099 |
| 1.3 | 0 | 0 | 2999 | 3099 | 2999 | 3099 |
| 1.35 | 0 | 0 | 2986.5 | 3099 | 2961.5 | 3124 |
| 1.4 | 0 | 0 | 2949 | 3132 | 2949 | 3149 |
| 1.45 | 0 | 0 | 2949 | 3149 | 2949 | 3149 |
| 1.5 | 0 | 0 | 2949 | 3149 | 2949 | 3149 |
| 1.55 | 0 | 0 | 2949 | 3149 | 2946.5 | 3149 |
| 1.6 | 0 | 0 | 2929 | 3149 | 2899 | 3174 |
| 1.65 | 0 | 0 | 2899 | 3189 | 2899 | 3199 |
| 1.7 | 0 | 0 | 2899 | 3199 | 2899 | 3199 |
| 1.75 | 0 | 0 | 2899 | 3199 | 2899 | 3199 |
| 1.8 | 0 | 0 | 2899 | 3199 | 2899 | 3199 |
| 1.85 | 0 | 0 | 2899 | 3199 | 2899 | 3209 |
| 1.9 | 0 | 0 | 2899 | 3214 | 2874 | 3236.5 |
| 1.95 | 0 | 0 | 2874 | 3239 | 2849 | 3249 |
| 2 | 0 | 0 | 2849 | 3249 | 2849 | 3249 |

gamma - log odds of differential assignment due to unobserved factors.

sig + - upper bound significance level.

sig- - lower bound significance level.

t-hat + - upper bound Hodges-Lehmann point estimate.

t-hat- - lower bound Hodges-Lehmann point estimate.

CI+ - upper bound confidence interval (a=.95).

CI- - lower bound confidence interval (a = .95).

Bangladesh (14%), and Nepal (11.3%) [78]. These observed disparities in the proportion of LBW might be attributed to variations in sample size, timing of data collection, and measurement methods, especially between self-reported birthweight and clinical records. From a policy perspective, these findings underscore the importance of strengthening maternal and neonatal health surveillance systems to ensure accurate data capture and comparability across regions. Moreover, the relatively lower LBW prevalence observed here may signal progress in maternal health service delivery, yet it also calls for targeted interventions to sustain and improve outcomes among vulnerable populations. Integrating LBW reduction strategies into regional health frameworks, especially those focused on equity and universal health coverage [79,80], can help address persistent gaps and ensure that improvements reach the most at-risk groups.

## Policy implications

The study's finding of a 10.42% LBW prevalence, notably below the global average of 14.7% indicates meaningful progress but also underscores the continued need to enhance maternal health services across West African countries. This progress may reflect growing alignment with the Global Nutrition Targets 2025, which aim to reduce LBW prevalence by 30% [39], and supports the broader objectives of Sustainable Development Goal (SDG) Target 3.2, which seeks to end preventable newborn deaths and lower neonatal mortality to at least 12 per 1,000 live births by 2030 [81].

The observed birthweight improvements associated with adequate ANC visits reaffirm the critical role of quality and timely maternal care in achieving optimal birth outcomes. Expanding access to comprehensive ANC, especially in line with the WHO-recommended ANC8 + contact model, presents a high-impact opportunity to further reduce LBW and ultimately advance maternal and neonatal health equity. Policymakers should prioritize interventions that strengthen ANC delivery systems, ensure early and comprehensive maternal monitoring, and address barriers to care among underserved populations to accelerate progress toward these global targets.

### Strengths and limitations

This study presents several methodological and contextual strengths. By employing PSM, the analysis reduces selection bias inherent to observational data and allows for stronger causal inferences regarding the impact of adequate antenatal care (ANC4+ and ANC8+) on birth weight outcomes. The operational definition of ANC adequacy aligns with WHO guidelines, and the stratified examination of ANC4+ versus ANC8 + enhances interpretability and policy relevance. Additionally, the use of robust treatment effect estimates (ATT and ATE) enables a nuanced understanding of ANC's effects across treated and total populations within diverse West African contexts. Overall, the study lies in the rigorous application of PSM complemented by thorough diagnostic analyses, which verified the robustness and quality of the matching process.

However, the study was not without limitations. Residual confounding from unobserved variables—such as nutritional status and environmental exposures—as the matching (PSM) was done for observable variables. Lastly, due to the cross-sectional nature of DHS data, the findings may be prone to social desirability and recall bias.

## Conclusion

This study found that adequate ANC (ANC4 or ANC8+) contacts have a positive effect on birthweight outcomes in West African countries, which underscores the importance of scaling up efforts to ensure comprehensive ANC coverage, especially ANC8 + . These findings highlight the potential adherence to comprehensive ANC strategies to reduce LBW prevalence and improve neonatal outcomes. Thus, stakeholders in the health sector should work on strengthening ANC through integrated nutrition support, infection screening, and psychosocial intervention to reduce the risk of LBW.

## Author contributions

**Conceptualization:** Aklilu Habte Hailegebireal.

**Formal analysis:** Aklilu Habte Hailegebireal.

**Investigation:** Angwach Abrham Asnake.

**Methodology:** Aklilu Habte Hailegebireal, Habtamu Mellie Bizuayehu, Samuel Hailegebreal Gele, Angwach Abrham Asnake.

**Software:** Aklilu Habte Hailegebireal, Samuel Hailegebreal Gele.

**Supervision:** Habtamu Mellie Bizuayehu.

**Writing – original draft:** Aklilu Habte Hailegebireal.

**Writing – review & editing:** Aklilu Habte Hailegebireal, Habtamu Mellie Bizuayehu, Samuel Hailegebreal Gele, Angwach Abrham Asnake.

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
