## [Decision Letter · Decision Letter 0]

6 Oct 2025

PGPH-D-25-02200

Boosting Birthweight through Better Prenatal Care in West African countries: insights from Propensity Score Matching

Dear Dr. Hailegebireal,

Thank you for submitting your manuscript to PLOS Global Public Health. After careful consideration, we feel that it has merit but does not fully meet PLOS Global Public Health’s publication criteria as it currently stands. Therefore, we invite you to submit a revised version (Minor comments) of the manuscript that addresses the points raised during the review process.

We look forward to receiving your revised manuscript.

Kind regards,

Shiyam Sunder, MBBS, MSc epidemiology, PhD

Academic Editor

Journal Requirements:

2. In the online submission form, you indicated that “The data for this study were obtained from the DHS program with a reasonable request. Thus, the one who needs the data supporting the findings of this study can get it in anonymised form from the DHS website at https://dhsprogram.com/Countries/ upon reasonable request in the same manner as the authors did.” 

3. Uploaded as supplementary information.

3. Please provide separate figure files in .tif or .eps format.

Reviewers' comments:

Reviewer's Responses to Questions

**Comments to the Author**

1. Does this manuscript meet PLOS Global Public Health’s publication criteria?

Reviewer #1: Partly

Reviewer #2: Yes

2. Has the statistical analysis been performed appropriately and rigorously?

Reviewer #1: Yes

Reviewer #2: I don't know

3. Have the authors made all data underlying the findings in their manuscript fully available (please refer to the Data Availability Statement at the start of the manuscript PDF file)?

Reviewer #1: No

Reviewer #2: Yes

4. Is the manuscript presented in an intelligible fashion and written in standard English?

Reviewer #1: No

Reviewer #2: Yes

Reviewer #1: The following are my observations

The manuscript is timely and explicitly reports important public health issue (low birth weight), with minor corrections I recommend it for publication as this will inform public health interventions in maternal and child healthcare.

The following are the areas to be addressed by authors; -

The manuscript partly meets PLOS Global Public Health’s publication criteria

The manuscript addresses a highly relevant public health issue (low birth weight in West Africa) using multi-country DHS data and propensity score matching (PSM), which is methodologically sound and ethically appropriate for secondary data use.

However: -

The framing in the Introduction could be more concise and explicitly linked to equity and global health relevance, as required by the journal.

The Discussion does not sufficiently contextualize findings within regional health policy frameworks or highlight implications for equity, health systems, and vulnerable populations.

Result sections are overly detailed with large tables that may overwhelm readers; clarity and focus on the most important findings would improve accessibility.

The statistical analysis been performed appropriately and rigorously

The use of PSM is appropriate for causal inference in observational DHS data. Matching diagnostics, sensitivity analyses (Rosenbaum bounds), and balance checks are clearly presented and robust. The statistical approach supports the conclusions.

Minor refinements may include the following: -

Reporting standardized effect sizes alongside mean differences.

Explicit mention of how missing data were handled.

A clearer justification for choosing caliper = 0.01 in radius matching.

The authors have not made all data underlying the findings fully available

The manuscript states that DHS data were used and available at https://dhsprogram.com. However, the Data Availability Statement does not currently comply with PLOS requirements: -

Authors must provide a direct DOI or persistent URL to the DHS datasets used, specifying country, year, and dataset type (IR file).

A statement such as: “The data are publicly available upon registration from the DHS Program (https://dhsprogram.com). The specific datasets used in this study include (list of 14 country-years, IR files).” should be added.

The manuscript is partially presented in an intelligible fashion and written in standard English

The manuscript is generally understandable but requires language polishing for clarity and conciseness.

There are grammatical inconsistencies (e.g., “Of the weighted sample, 71.38% and 14.54% of women received ANC4+ and ANC8+, respectively. All matching diagnostics demonstrated strong covariate balance and confirmed the validity of the treatment effect estimates. The treatment and control groups were well comparable…” the statements should be streamlined).

Some sections are repetitive (e.g., ANC definitions repeated in Introduction, Methods, and Results).

Long tables could be simplified or moved to supplementary materials, with summary results in the main text, to enhance readability and comprehension.

Reviewer #2: To the editorial team, thank you for the opportunity to review this manuscript. This is an important research piece that has potentially a high impact on maternal and child health. However, I have a few comments that can help make it stronger.

1. in the methods sections, the selection of the countries and the specific years in review should be explained methodically, or in case of convenience, should be indicated.

2. the use of the codes- 9996, 9997, etc, should be explained in more detailed.

3. Since the data is from DHS - a secondary data, I am more concerned with data quality issues rather than recall bias and social desirability effect. It will be helpful to explain how that could be a concerned, in the limitation section.

4. I will encourage to review the references.

5. This is actually a well written manuscript; however, I will encourage a review for typos/grammar. In the results section, please correct the typos in the first paragraph (6th line).

**Do you want your identity to be public for this peer review?** For information about this choice, including consent withdrawal, please see our Privacy Policy

Reviewer #1: **Yes: ** Emmanuel Ekung

Reviewer #2: No

---

## [Editor Report · Decision Letter 1]

6 Nov 2025

Enhancing birth weight outcomes through improved antenatal care in West African countries: Evidence from Propensity Score Matching Analysis

PGPH-D-25-02200R1

Dear Mr Hailegebireal,

We are pleased to inform you that your manuscript 'Enhancing birth weight outcomes through improved antenatal care in West African countries: Evidence from Propensity Score Matching Analysis' has been provisionally accepted for publication in PLOS Global Public Health.

Best regards,

Shiyam Sunder, MBBS, MSc epidemiology, PhD

Academic Editor

All comments are very well addressed.